# Use of Intrinsic Entropy to Assess the Instantaneous Complexity of Thoracoabdominal Movement Patterns to Indicate the Effect of the Iso-Volume Maneuver Trial on the Performance of the Step Test

**DOI:** 10.3390/e26010027

**Published:** 2023-12-26

**Authors:** Po-Hsun Huang, Tzu-Chien Hsiao

**Affiliations:** 1Institute of Computer Science and Engineering, College of Computer Science, National Yang Ming Chiao Tung University, Hsinchu 30010, Taiwan; pohsun.cs05@nycu.edu.tw; 2Department of Computer Science, College of Computer Science, National Yang Ming Chiao Tung University, Hsinchu 30010, Taiwan; 3Institute of Biomedical Engineering, College of Electrical and Computer Engineering, National Yang Ming Chiao Tung University, Hsinchu 30010, Taiwan

**Keywords:** intrinsic entropy, thoracoabdominal movement, respiratory inductance plethysmography

## Abstract

The recent surge in interest surrounds the analysis of physiological signals with a non-linear dynamic approach. The measurement of entropy serves as a renowned method for indicating the complexity of a signal. However, there is a dearth of research concerning the non-linear dynamic analysis of respiratory signals. Therefore, this study employs a novel method known as intrinsic entropy (IE) to assess the short-term dynamic changes in thoracoabdominal movement patterns, as measured by respiratory inductance plethysmography (RIP), during various states such as resting, step test, recovery, and iso-volume maneuver (IVM) trials. The findings reveal a decrease in IE of thoracic wall movement (TWM) and an increase in IE of abdominal wall movement (AWM) following the IVM trial. This suggests that AWM may dominate the breathing exercise after the IVM trial. Moreover, due to the high temporal resolution of IE, it proves to be a suitable measure for assessing the complexity of thoracoabdominal movement patterns under non-stationary states such as the step test and recovery. The results also demonstrate that the instantaneous complexity of TWM and AWM can effectively capture instantaneous changes during non-stationary states, which may prove valuable in understanding the respiratory mechanism for healthcare purposes in daily life.

## 1. Introduction

In 1948, Shannon extended the concepts of Hartley function and Boltzmann entropy to propose an important idea called Shannon entropy in information theory [1]. For a system *X* that contains *N* states, the Shannon entropy is defined as follows:(1)$$H\left(X\right)=-\sum_{i=1}^{N}p_{i}\log_{2}p_{i}$$
where $p_{i}$ is the probability that ith state would occur. Therefore, while all states have the same probability, the Shannon entropy would have the maximum value, which is equivalent to the Hartley function. In other words, the Shannon entropy gives the average amount of information that each state of the system can represent. Alongside Shannon entropy, which quantifies information, Kolmogorov and Sinai also introduced a measure to assess the rate of information generation, which is called Kolmogorov-Sinai entropy or Kolmogorov entropy [2,3]. The Kolmogorov entropy calculates the difference entropy value of consecutive processes. It can be easily represented as
(2)$$H_{KS}=\underset{N\to \infty }{\lim}\frac{1}{N}\sum_{n=0}^{N-1}H_{n+1}-H_{n}$$
where $H_{n}$ denotes the entropy value of the nth time or dimension. In addition to information theory, the concepts of Shannon entropy and Kolmogorov entropy are also famous for measuring the complexity or randomness of time series. Recently, there has been a growing interest in the non-linear dynamics of physiological signals. Numerous algorithms based on Shannon entropy or Kolmogorov entropy were proposed to measure the complexity or randomness of complex systems [4].

Approximate entropy (ApEn) and sample entropy (SampEn) are the most well-known methods to indicate the irregularity of time series [5,6]. Additionally, many algorithms related to the aforementioned entropy methods were also proposed to measure the complexity of physiological signals, including electrocardiography (ECG) [7,8,9,10,11,12,13], photoplethysmography (PPG) [14,15,16], impedance cardiogram (ICG) [17], electroencephalography (EEG) [18,19,20,21,22], etc., under different conditions. 

In addition to the responses of the heart (indicated by ECG or PPG) and brain (indicated by EEG), the activity of the lungs, especially breathing, is one of the most important mechanisms for sustaining human life. To perform breathing, the movement of the thoracic and abdominal walls is needed. The previous studies pointed out that the asynchrony between thoracic wall movement (TWM) and abdominal wall movement (AWM) is related to the health of humans [23,24,25,26,27]. The TWM and AWM can be indicated by respiratory inductance plethysmography (RIP). However, the variation of TWM and AWM was less noted since the temporal resolution of respiratory rate is limited [28,29]. Some studies apply complexity analysis to RIP to indicate the difference among sleep stages [30] and to quantify ventilation variability [31]. A study also computed the ApEn of the respiratory signal during exercise to detect the breathing pattern disorder [32]. However, fewer studies discuss the effect of TWM and AWM during exercise using non-linear dynamic metrics.

In this study, to address the temporal resolution problem of respiratory rate, intrinsic entropy (IE) is used to analyze the complexity of the RIP signal during resting state, exercise state, recovery state, and iso-volume maneuver (IVM) trial state. Unlike the aforementioned entropy method, IE is able to measure the instantaneous complexity of time series, which means that the setting of a time window is not necessary [33]. Therefore, the IE is suitable for analyzing physiological signals, which are usually non-stationary.

## 2. Materials and Methods

### 2.1. Intrinsic Entropy (IE)

The IE is calculated by the following procedure [33]: First, the famous decomposition method empirical mode decomposition (EMD) is applied to decompose the signal into intrinsic mode functions (IMFs), which meet two conditions: (1) the mean envelope is close to zero; and (2) the numbers of the zero-cross and extremum at most differ by 1 [34]. Second, the AM-FM decomposition is applied to extract the instantaneous amplitude [35]. Last, calculate the probability and entropy value by Equation (1).

In the first step, the classic EMD is described as follows: Consider an input signal $X_{raw}(t)$, Let $x\left(t\right)=X_{raw}(t)$, find the local maximum and local minimum of $x\left(t\right)\;$ to generate upper envelope U(t) and lower envelope L(t) by cubic spline. Calculate the mean envelope $M\left(t\right)$
(3)$$M\left(t\right)=\frac{U\left(t\right)+L(t)}{2}$$

Let the output $h\left(t\right)=x\left(t\right)-M(t)$, check whether the output $h\left(t\right)$ meets the condition of the IMF or not. If $h\left(t\right)$ is not IMF, let $x\left(t\right)=h\left(t\right)$ to redo above procedure; else output $h\left(t\right)$ as an IMF and check the residual $r\left(t\right)=x\left(t\right)-h\left(t\right)$ is mono or not. If the residual $r\left(t\right)$ is mono, end the EMD; else, let $x\left(t\right)=r\left(t\right)$ to redo the above procedure. In this study, a modified version based on the Gram–Schmidt process was used for better orthogonality between $h\left(t\right)$ and $M\left(t\right)$. For convenience, let vector $\overset{⃑}{m},and\;\overset{⃑}{x}$ denote $M\left(t\right)\;and\;x\left(t\right)$, respectively. The procedure $h\left(t\right)=x\left(t\right)-M(t)$ is replaced with $h\left(t\right)=x\left(t\right)-M^{\prime }\left(t\right)$ where $M^{\prime }\left(t\right)$ is
(4)$$M^{\prime }\left(t\right)=\frac{\overset{⃑}{x}\cdot \overset{⃑}{m}}{{\left\|\overset{⃑}{m}\right\|}_{2}^{2}}\times M(t)$$
where · is inner product and ${\left\|\overset{⃑}{m}\right\|}_{2}^{2}$ is $\sum M^{2}(i)$.

Thus, if N IMFs are extracted, the relation between the input signal, $X_{raw}(t)$ and IMFs is
(5)$$X_{raw}\left(t\right)=\sum_{i=1}^{N}{IMF}_{i}(t)+r(t)$$

After the IMFs are extracted, the AM-FM decomposition is applied to extract the instantaneous amplitude [36]. Though the instantaneous amplitude can be easily calculated with the Hilbert transform as
(6)$$a_{k}\left(t\right)=\sqrt{{IMF}_{k}^{2}\left(t\right)+{\left\{Hb[{IMF}_{k}\left(t\right)]\right\}}^{2}}$$
where $a_{k}\left(t\right)$ is the instantaneous amplitude at time *t* of kth IMF; Hb is the Hilbert transform. The result of $a_{k}$ estimated from the above equation is not smooth enough for further calculation [35]. Thus, the AM-FM decomposition was applied in this study. The procedure for AM-FM decomposition involves the following steps:
Initialize the iterator j.Calculate the upper envelope $e_{j}$ by interpolating the local maximum of |${IMF}_{k}$|.For all t, if there exists at least one *t* that $\frac{{IMF}_{k}(t)}{e_{j}\left(t\right)}$ larger than 1, let ${IMF}_{k}=\frac{{IMF}_{k}}{e_{j}},j=j+1$ to redo II.Else, output $a_{k}=\prod_{i=1}^{j}e_{i}$After the $a_{k}$ is extracted, a power distribution at time *t* can be obtained as $\left[a_{1}\left(t\right),\;a_{2}\left(t\right),\ldots ,a_{N}(t)\right]$. Therefore, the probability can be calculated as
(7)$$p_{i}(t)=\frac{a_{i}(t)}{\sum_{k=1}^{N}a_{k}(t)}$$The entropy value at time *t* is
(8)$$IE\left(t\right)=-\sum_{i=1}^{N}p_{i}(t)\log_{2}p_{i}(t)$$Dividing $\log_{2}N$ is able to obtain the normalized IE.

### 2.2. Experiment Description

The experiment was conducted at National Chiao Tung University and was approved by the Research Ethics Committee for Human Subject Protection of National Chiao Tung University (NCTU-REC-106-052). A total of 27 participants were recruited for this experiment. All the data and signals of participants have been de-identified. Eight participants’ data were excluded in this study because of serious signal distortion and data loss.

The experimental design consisted of two phases, each with three stages: rest, step test, and recovery. The two phases were connected by an IVM trial. Participants sat in a chair for 5 min in the resting state to acquire a signal. Then, they performed a 3 min step test based on the standard procedure established by the Sports Administration of the Ministry of Education in Taiwan [36]. After exercise, participants rested in a chair for 5 min to recover. Before the second phase, participants performed an IVM trial for 5 min. After completing the IVM trial, participants repeated the same stages as in the first phase. A respiratory inductance plethysmography (RIP) device (Ambu Sleepmate Ripmate Inductance Belt Thorax, AMBU, Columbia, MD, USA) and an electrocardiography (ECG) device (BEST-C-04056, BioSenseTek, Taipei, Taiwan) were used in this experiment to acquire the signal during 2 phases and the IVM trial. The sampling rate for both RIP and ECG was 250 Hz.

#### 2.2.1. Step Test

In this study, the step test followed the protocol established by the Sports Administration of the Ministry of Education in Taiwan [36]. The height of the step was 35 cm; the frequency of stepping was 24 steps per minute with a total of 3 min; and the tempo of stepping was 4 beats per step. On the first beat, the participants stepped onto the bench with their left foot; on the second beat, the participants stepped onto the bench with their right foot; on the third beat, the participants stepped off the bench with their left foot and onto the ground; on the fourth bench, the participants stepped off the bench with their right foot and onto the ground. Hence, one cycle for a step was 2.5 s. The step test is useful to increase the participants’ heart rate in a short period of time [37]. In addition, cardiopulmonary function can also be quantified by an index named the physical fitness index (PFI), which is calculated as follows:(9)$$PFI=\frac{D}{2\times (B_{1}+B_{2}+B_{3})}$$
where *D* is the duration (second) of the step test; $B_{1}$, $B_{2}$, and $B_{3}$ are the numbers of heart beat between the 1 and 1.5 min, 2 and 2.5 min, and 3 and 3.5 min after step test, respectively. Some studies showed that PFI is useful to indicate the participants’ physical ability [37].

#### 2.2.2. Iso-Volume Maneuver (IVM) Trial

The IVM trial in this study followed the protocol established by [38]. The purpose of the IVM trial was to help the participants with abdominal breathing. There are 3 steps per minute for a total of 5 min. The first step of the IVM trial is spontaneous breathing. The participant would be asked to perform spontaneous breathing for 24 s. Secondly, the participants inhaled deeply for 6 s. Lastly, the participants held their breath for 30 s and tried to make their abdomen concave and convex, following the instructions on the computer monitor [38].

### 2.3. Statistical Analysis

For the purpose of showing the statistical analysis, quantifying the value of the IE series is needed. Because the effect of TWM and AWM is the main objective of this study, we quantified the IE values by calculating the difference in the area under the curve (dAUC) between the IE of TWM and the IE of AWM.
(10)$$dAUC=\sum_{t=1}^{N}{IE}_{TWM}\left(t\right)-\sum_{t=1}^{N}{IE}_{AWM}(t)$$

Thus, the smaller value indicates the greater influence of AWM.

To indicate the effect of the IVM trial, the paired *t*-test was applied in this study. The paired *t*-test compares the difference between before and after the IVM trial in each stage. The Mann–Whitney U test was used to compare the dAUC between the group with a PFI increase or unchanged and the group with a PFI decrease in the same stage. In addition, the Fisher’s Exact Test, an alternative method of the Chi-Squared Test for small sample sizes, was used to indicate whether the PFI change influenced the difference in dAUC between the IE of TWM and AWM. 

## 3. Results

Figure 1 shows the RIP signal, corresponding IE series, and heart rate (HR) of a total of 19 subjects during the IVM trial. In Figure 1a, the RIP signal can indicate the fluctuation in the 3 stages of the IVM trial. Figure 1b showed that the IE of TWM and AWM decreased during breath holding compared with spontaneous breathing. In addition, the IE of AWM would be a little larger than the IE of TWM during this stage. Moreover, the fluctuation of IE in TWM and AWM was similar to the HR (Figure 1c) during the whole IVM trial.

Figure 2 indicates the influence of the IVM trial. Figure 2a revealed that the HR after the IVM trial during resting was a little bit larger than the HR before the IVM trial. However, the HR did not show a significant difference between before and after the IVM trial during the step test and recovery. During rest, the IE of AWM was similar to the IE of TWM (Figure 2b) before the IVM trial, but after the IVM trial, the IE of AWM was larger than the IE of TWM. During the step test, compared with before the IVM trial, the IE of AWM was larger than the IE of TWM for a longer time from the exercise start (see the orange line in Figure 2b,c during the step test). During recovery, compared with before the IVM trial, the IE of AWM was smaller than the IE of TWM only for a short time from the start of recovery (see the orange line in Figure 2b,c during recovery).

Figure 3 is to understand the effect of physical endurance; the 19 subjects were classified into two groups according to the PFI value calculated from the recovery stage. The total of 9 subjects’ PFI increased or remained unchanged after the second step test. Otherwise, the total of 10 subjects’ PFI decreased compared with the first-time step test before the IVM trial. For the group with increased or unchanged PFI, there was no significant change in HR in the rest, step test, or recovery before the IVM trial compared with after the IVM trial. In contrast, the HR increased significantly after the IVM trial in the group, with a PFI decrease in the rest and step tests. Since the PFI was calculated from the heartbeat in recovery, the HR of the group with PFI decreased slower after the IVM trial compared with before the IVM trial.

Figure 3b,c indicate the influence of the IVM trial on the IE of AWM and TWM. Though the trends of IE of AWM and TWM before and after the IVM trial were similar to those in Figure 2b,c where IE of AWM tends to be larger than IE of TWM, the influence of the IVM trial seems to be weaker in the group with a PFI increase or unchanged than the group with a PFI decrease. During rest, the IE of AWM was larger than the IE of TWM after the IVM trial compared with before the IVM trial in both groups (the double-headed arrow in Figure 3). During the step test, the timing of IE of AWM being smaller than IE of TWM was later in both groups after the IVM trial, but this effect is not significant (the moving of the orange line is short in Figure 3b,c during the step test). During recovery, the timing of IE of AWM being smaller than IE of TWM was earlier in both groups after the IVM trial; this effect in the group with a PFI decrease is stronger than that in the group with a PFI increase or unchanged (the moving of the orange line is longer in the group with a PFI decrease compared with the group with a PFI increase or unchanged).

Table 1 shows the dAUC in each stage before and after the IVM trial. The dAUC declined in rest and recovery significantly after the IVM trial. The dAUC increased in step but was not significant.

Figure 4 represents the result of the difference in dAUC between before and after the IVM trial in the two groups. Though the difference in dAUC in the group with a PFI decrease was smaller than the group with a PFI increase of unchanged in each stage, no statistically significant differences were found.

Table 2, Table 3 and Table 4 showed the contingency table for the Fisher’s Exact Test. The dAUC decreased in both groups during the three stages. No significant differences were found in these three tables.

## 4. Discussion

We present a methodology to demonstrate the effect of the IVM trial on respiratory exercise. It is highly likely that IVM trials will have a profound impact on increasing the complexity of the AWM while simultaneously reducing the complexity of the TWM. The results displayed in Figure 1 illustrate that the IE of AWM is expected to be greater than the IE of TWM during breath-holding. Following the IVM trial, Figure 2 and Figure 3 demonstrate a significant decrease in the IE of TWM, resulting in the IE of AWM surpassing that of TWM. Is it reasonable to consider the IVM trial as a means to enhance abdominal utilization for breathing? From our perspective, given that the IE value would be higher when more IMFs with comparable power are present, the answer to this question is affirmative. Prior to the IVM trial, the IE of TWM mirrored that of AWM, suggesting a mutual regulation between the two. However, after the IVM trial, the IE of TWM decreased, indicating that the respiratory exercise was dominated by AWM. This can be attributed to the fact that the abdominal muscles drive the chest muscles, potentially resulting in the RIP of AWM containing multiple dominant IMFs to elevate the IE value. These circumstances are clearly evident during the rest and recovery stages, where the dAUC significantly decreases (Table 1). Nevertheless, during the step test stage, the IE of AWM decreased due to the relative difficulty of maintaining abdominal breathing. Figure 3 provides evidence that the impact of the IVM trial on the complexity of TWM and AWM is more pronounced in the group with a decrease in PFI, indicating poorer physical fitness. Individuals with superior physical fitness typically excel at regulating their breathing. Conversely, for those with lower physical fitness, the IVM trial may assist in regulating their breathing in order to successfully complete high-intensity exercise for a second time. However, though in Figure 4, the group with a PFI decrease had a lower value in the difference of dAUC between before and after the IVM trial than the group with a PFI increase, no significant differences were found by the Mann–Whitney U test. Additionally, no significant difference was found in Table 2, Table 3 and Table 4 by Fisher’s Exact Test, which means the change in PFI may not be directly affected by the change in dAUC. Nonetheless, the dAUC cannot directly reflect the timing of the change in the IE of AWM from smaller to greater than the IE of TWM. How to analyze the cross-point by statistic still needs further study.

The initial IVM technique, which solely encompasses the concepts of belly-in and belly-out, was utilized for the purpose of calibrating the RIP [39]. Furthermore, the IVM also aids individuals in directing their focus towards the movement of the abdomen, thereby facilitating continuous oversight of the inward and outward motion of the abdominal wall through the activation and relaxation of the abdominal muscles [40]. In order to assist users in achieving abdominal breathing, Chen devised the IVM trial [38]. Throughout the IVM trial, the users’ instantaneous phase difference between the TWM and the AWM is heightened, with variations being observed during different breathing patterns through the measurement of the asynchrony of the thoracoabdominal movement [38]. Furthermore, a study has also demonstrated the manner in which the IVM trial augments the phase difference between TWM and AWM, consequently influencing the performance of the step test [41]. The assessment of the asynchrony of the thoracoabdominal movement holds considerable significance within the clinical domain. Numerous research endeavors have substantiated the correlation between the asynchrony of the thoracoabdominal movement and various physical, psychological, or respiratory muscle-related ailments [23,24,25,26,27,42,43]. However, the precise relationship between the asynchrony of the thoracoabdominal movement and the complexity of the thoracoabdominal movement has yet to be definitively established. Although certain studies suggest a potential connection between respiratory rate variability or ventilation and complexity measurements [30,31,44], it remains imperative to conduct further investigations in order to fully comprehend the underlying implications of the complexity of the thoracoabdominal movement. In addition to the exercise, the asynchrony of the thoracoabdominal movement has been an important index for clinical outcomes such as successful extubation prediction [42] and chronic obstructive pulmonary disease patients’ discrimination [43]. The IE of the thoracoabdominal movement may also provide useful information for clinical use, including diagnosis or rehabilitation.

In addition, though in Figure 1, during the IVM trial, the average IE had the trend that the IE would be larger during spontaneous breathing, decrease during inhaling deeply, and the IE of AWM had a larger value than the IE of TWM during holding breathing, we need to acknowledge that not all subjects had this trend. Four interest cases were selected, for example. Figure 5a shows a case that did not have a similar trend compared with Figure 1b. In this subject, the IE of AWM dropped significantly during the early IVM trial. However, during the later IVM trial (after 150 s), the IE of AWM returned to be larger than the IE of TWM. This situation may be due to the fact that the subject was not familiar with the procedure of the IVM trial. After becoming familiar with this subject, a trend emerged. For Figure 5b, most of the time during the IVM trial, the IE of AWM was larger than the IE of TWM. It is possible that the subject is already used to breathing through the abdomen. In Figure 5c, although the IE decreased while holding the breath, which is similar to Figure 1b, the IE of AWM was not greater than the IE of TWM in the IVM trial. This may have been caused by a training failure. The subject had not learned how to breathe with the abdomen. Figure 5d is a standard example of a trend similar to the one in Figure 1b. Though these inferences still need to be verified, the IE provides valuable insights into the use of respiratory muscles. Furthermore, the IE is available to indicate very high temporal resolution; it is suitable to apply in precision medicine to customize personal health care guidelines. 

Since the IE has the particular property that it does not need to set the size of the time window or the ratio of overlapping for entropy computation, directly comparing the result of the IE with other conventional entropy methods is not appropriate. Nonetheless, we computed some conventional entropy of TWM and AWM during IVM for reference. Note that since the time complexity of SampEn is too large, the permutation entropy (PE) [45] and dispersion entropy (DE) [18] were applied here. The size of the time window was 625 points, which was 2.5 s. The overlapping ratio was set to 0%. Figure 6 indicates the result. To indicate the effect of TWM and AWM in three steps during the IVM trial by PE and DE is not significant, which may be caused by the inappropriate setting of the size of the time window and the ratio of overlapping. The setting of these two parameters is usually a critical issue in conventional entropy methods. In contrast, the IE does not need to set these two parameters. This property makes IE useful to indicate the intrinsic information within a time series. 

Some studies are concerned that the entropy computation would be affected by non-stationarity [46]. However, since the IE applied EMD, which is a popular method to deal with non-stationary signals, the result of the IE would not be affected by non-stationarity. The EMD is useful to eliminate the trend within the time series [47]. For the IE computation, the trend component would not be included. All IMFs extracted by EMD for IE computation are stationary. The IE is directly affected by the intrinsic fluctuation of the raw signal instead of non-stationarities. More detail about IE can be referred to in [33], which contains more synthetic signals to indicate how IE measures instantaneous complexity. Additionally, the value of IE is independent of the amplitude of the raw signal. Recalling Equation (5), we can find that the IMFs will be scaled proportionally, either amplified or reduced, based on the amplitude of the original signal. However, after substituting into Equation (7), the influence of amplitude would disappear.

This study contains some limitations. First, more samples are needed to verify the influence of the IVM trial. Additionally, the control group is also necessary for further comparison. Secondly, calculating the variance of the raw TWM and AWM signals may indicate the influence of the IVM trial (Figure 7), which implies a similar conclusion as IE did. Though the patterns of these two metrics were different, understanding the relationship between variance and IE is needed for further study. The third is to indicate the interaction between TWM and AWM; cross entropy or transfer entropy may be a more appropriate method. Previous studies applied cross entropy or transfer entropy to indicate the interaction between different physiological signals [6,48,49,50,51,52,53,54,55]. Therefore, the cross-IE needs to be developed. The third one is whether the PFI is a suitable index to indicate physical endurance. Most studies would also measure oxygen consumption (VO_2_) to indicate exercise capacity [56,57,58,59]. Measurement of this index and related ventilation indices is needed in future studies for comparison.

## 5. Conclusions

In this study, the effect of the IVM trial was indicated by the complexity analysis performed by IE, which showed a very short timescale dynamic of TWM and AWM. The result revealed that the IE of TWM tended to decrease and the IE of AWM tended to increase after the IVM trial. Since the movement of the thoracoabdominal is an important index to indicate the respiratory mechanism, measuring its complexity continually by IE would be useful for healthcare.

## Figures and Tables

**Figure 1 entropy-26-00027-f001:**
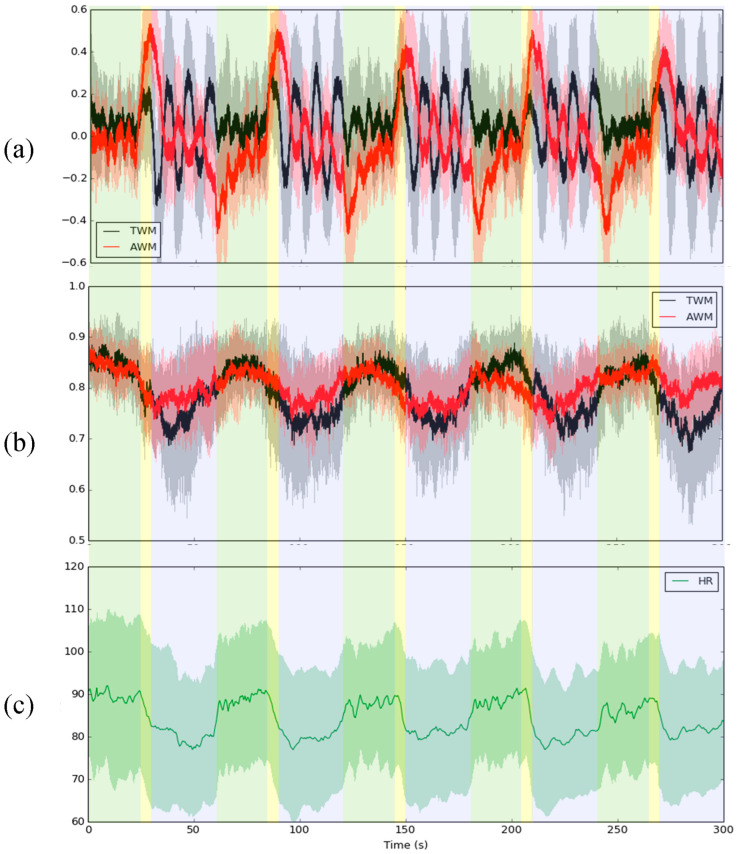
The average (**a**) RIP, (**b**) IE of TWM (black solid line) and AWM (red solid line), and (**c**) HR of a total of 19 subjects during the IVM trial The shaded area means one standard deviation from the mean of all subjects. The green, yellow, and blue block areas mean 24 s spontaneous breathing, 6 s inhale, and 30 s hold breathing with abdomen concave and abdomen convex, respectively.

**Figure 2 entropy-26-00027-f002:**
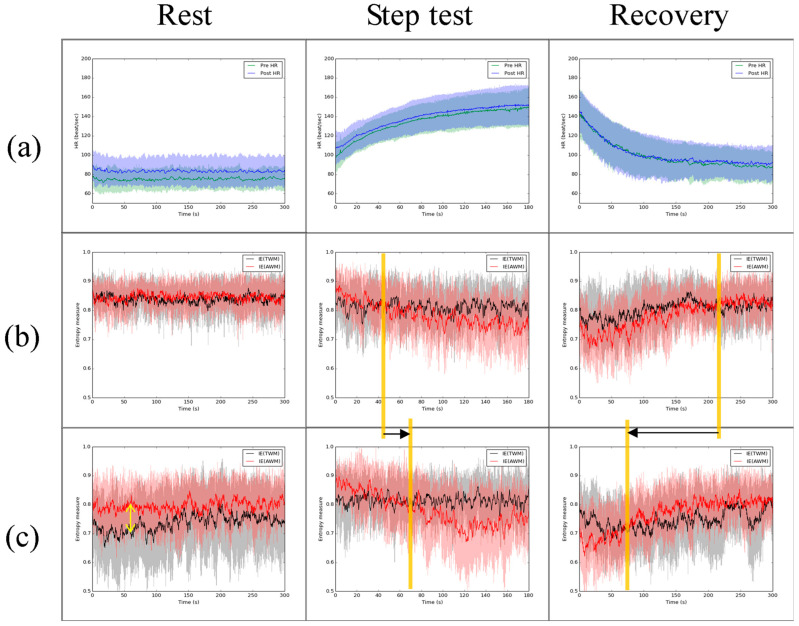
The comparison of HR and IE of TWM (black line) and AWM (red line) during each stage before and after the IVM trial. (**a**) HR (green line: before IVM; blue line: after IVM), (**b**) the IE before IVM, and (**c**) the IE after IVM. The yellow arrow shows the difference between IE of TWM and IE of AWM after IVM. The orange vertical line shows the point in time when the IE of AWM changed from larger (smaller) to smaller (larger) than the IE of TWM and black arrows indicate the movement of orange vertical line before and after IVM. The black and red lines are moving averages of the series of the mean IE. The shaded area means one standard deviation from the mean of all subjects.

**Figure 3 entropy-26-00027-f003:**
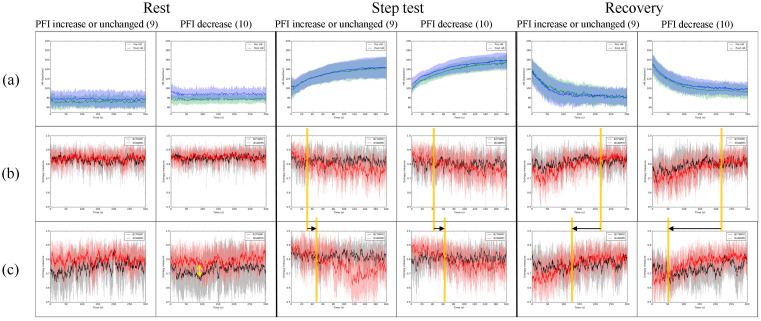
The group with PFI increased or decreased versus the group with PFI decreased in (**a**) HR and IE of TWM and AWM of (**b**) before the IVM trial and (**c**) after the IVM trial. Green solid line: the average HR before the IVM trial; blue solid line: the average HR after the IVM trial; black solid line: the moving average of the average IE of TWM; red solid line: the moving average of the average IE of AWM. The shaded area indicates one standard deviation from the solid line. The orange vertical line shows the point in time when the IE of AWM changed from larger (smaller) to smaller (larger) than the IE of TWM; black arrow: the movement of orange vertical line before and after IVM; yellow arrow: the difference between IE of TWM and IE of AWM.

**Figure 4 entropy-26-00027-f004:**
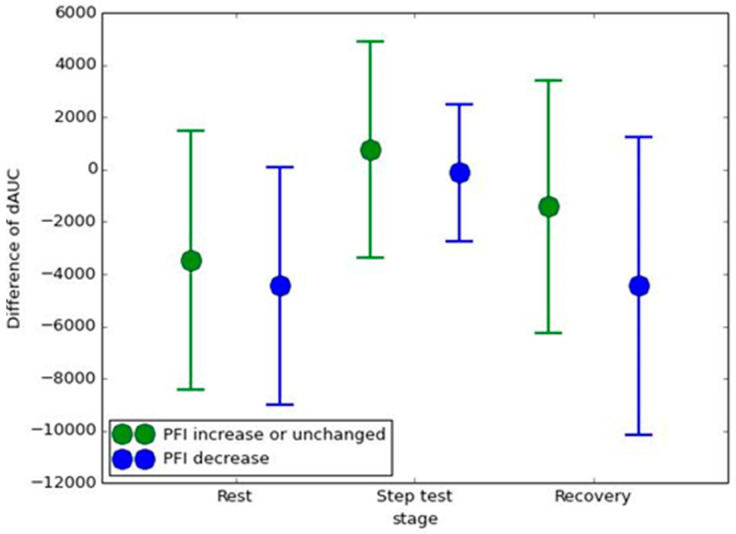
The difference in dAUC between before and after the IVM trial in the group with PFI increased or remained unchanged, and in the group with PFI decreased.

**Figure 5 entropy-26-00027-f005:**
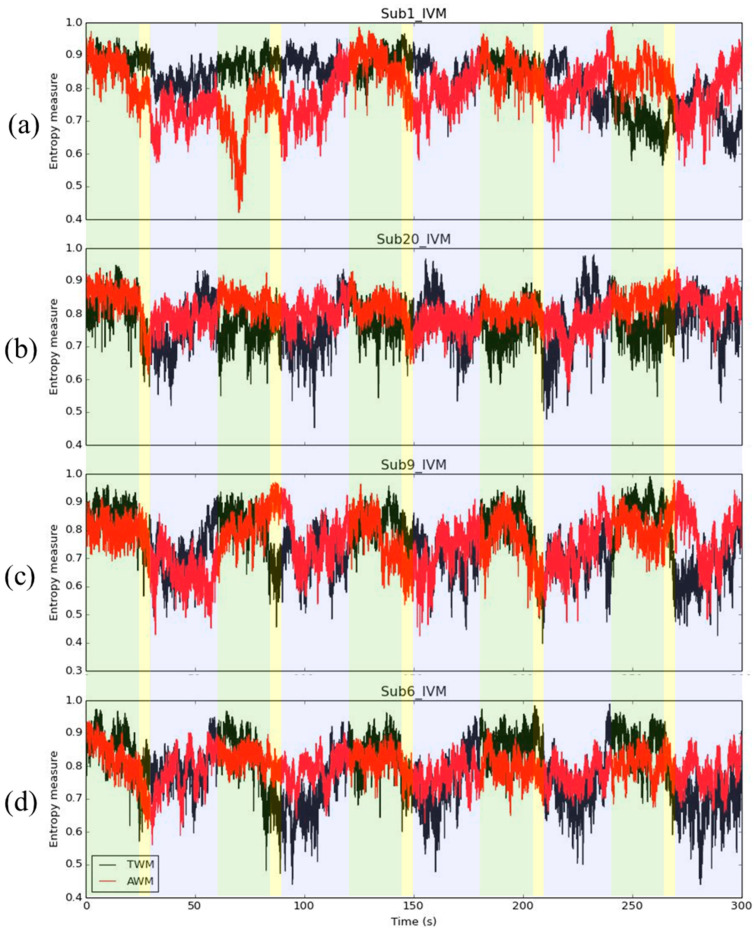
The case of IE during IVM trials. For the description of (**a**–**d**), see context.

**Figure 6 entropy-26-00027-f006:**
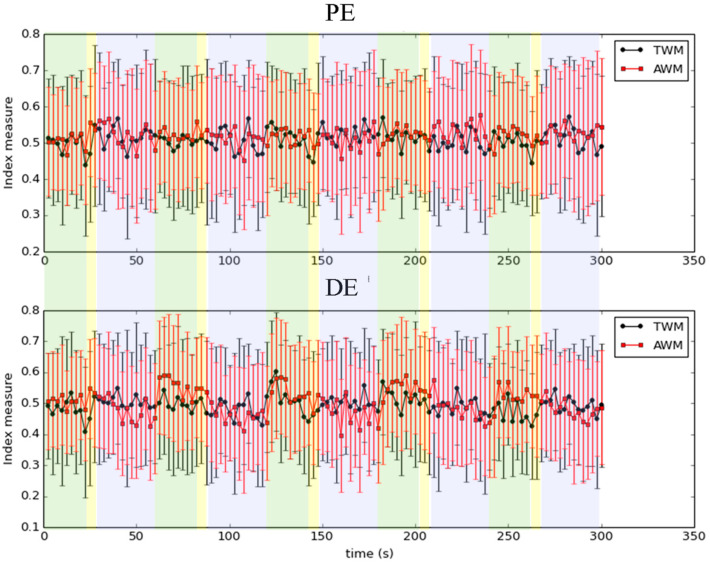
The irregularity of TWM and AWM was indicated by PE and DE during the IVM trial. The error bar indicates one standard deviation. The meaning of each colored block is the same as in Figure 1.

**Figure 7 entropy-26-00027-f007:**
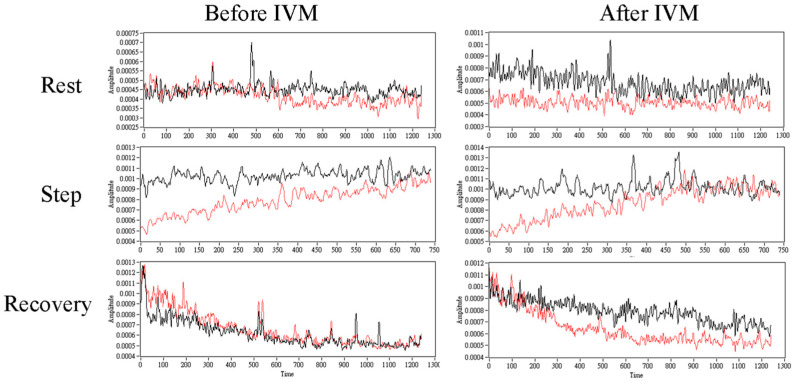
The comparison of the standard deviation of RIP for TWM (black line) and AWM (red line) during each stage before and after the IVM trial. The time window size was set to 625 points, which was 2.5 s, and an overlapping ratio was set to 90.04%, which was 565 points.

**Table 1 entropy-26-00027-t001:** The dAUC of all subjects in each stage before and after the IVM trial.

ALL Subjects	Rest	Step	Recovery
Before IVM	−499.15 ± 2053.6	1141.74 ± 2969.58	1596.73 ± 3046.28
After IVM	−4475.17 ± 6182.31 **	1434.31 ± 3856.01	−1414.40 ± 4799.54 *

** *p*-value < 0.001; * *p*-value < 0.05.

**Table 2 entropy-26-00027-t002:** The contingency table, according to the dAUC, changed in each group during rest.

Rest	PFI Increase or Unchanged	PFI Decrease
dAUC increase	1	2
dAUC decrease	8	8

**Table 3 entropy-26-00027-t003:** The contingency table, according to the dAUC, changed in each group during the step test.

Step	PFI Increase or Unchanged	PFI Decrease
dAUC increase	3	4
dAUC decrease	6	6

**Table 4 entropy-26-00027-t004:** The contingency table, according to the dAUC, changed in each group during recovery.

Recovery	PFI Increase or Unchanged	PFI Decrease
dAUC increase	4	2
dAUC decrease	5	8

## Data Availability

The data presented in this study are available on request from the corresponding author. The data are not publicly available due to the privacy of the subjects.

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
