# Peer review of "Use of Intrinsic Entropy to Assess the Instantaneous Complexity of Thoracoabdominal Movement Patterns to Indicate the Effect of the Iso-Volume Maneuver Trial on the Performance of the Step Test"

_entropy, 2023, doi:10.3390/e26010027_

Round 1
Reviewer 1 Report
Comments and Suggestions for Authors
The study applied a method validated in ref #33 for the assessment of irregularity of thoracoabdominal movement patterns after iso-volume maneuver trial during step test.
The study is of interest and contains original data. However, its methodological rigor is weak, and the added value of the approach compared to the huge mass of tools assessing irregularity present in literature was not proven.
1) Results are not tested again different methods able to assess irregularity. The authors must compare their conclusions with more popular methods present in literature for the assessment of irregularity of time series.
2) The authors must review more carefully the field of the computation of entropy over short time series. Some suggestions are provided above. Remarkably, some of the abovementioned articles deal with the analysis of the respiratory signal and physical exercise. Introduction must be made more insightful to avoid the impression that the applied method is the sole technique that can be applied.
3) Entropy computation requires stationarity, but the authors did not test this prerequisite. Please test the stationarity of the mean and variance (at the minimum) before applying your tools (see e.g. V. Magagnin et al, Physiol Meas, 32, 1775-1786, 2011) in order to prove that the conclusions are not the trivial effect of the presence of nonstationarities and have nothing to do with irregularity of the time series. This is mandatory because the exploited protocol evoked important nonstationarities as evident from the figures.
4) The study is extremely weak from a statistical standpoint. Indeed, it features three factors, namely experimental condition, training condition and time. Please demonstrate that the study has a sufficient power to provide robust conclusions. Please report the statistical analysis session in which this issue is deeply faced (with appropriate statistical tests) as well as the issue of the multiple comparisons.
5) The adjective “instantaneous” is unjustified given that entropy was computed over a given frame length. Please report the frame length and prove that the conclusions are stable while varying the frame length, namely while decreasing it towards a situation of “instantaneousness”.
6) It is unclear whether the method is independent of the amplitude of the signal. Please show that adjusting variance of the signal by multiplying the deviation from the mean by a constant did not vary the conclusions.
7) My personal impression is that the trivial analysis of the variance of the time series, or part of it (i.e. modes), provides similar results. Please report the variance and provide statistical analysis within a bar graph.
Comments on the Quality of English Language
Minor editing is necessary.
Reviewer 2 Report
Comments and Suggestions for Authors
The paper reports of the effectiveness of employing intrinsic entropy (IE) of the respiratory inductance plethysmography (RIP) to measure short-term changes in thoracic wall movement (TWM) and abdominal wall movement (AWM) movement patterns during resting, step test, recovery, and iso-volume maneuver.
The paper is well written, the methods are well described, and the topic is interesting. In my opinion, only a few concerns need to be addressed before publication:
1) In my opinion employing a statistical analysis to compare the IE during the different phases and from TWM and AWM. For instance, the Authors could use a 2-way ANOVA including both the phases and the anatomical location. Moreover, a multiple comparison should be performed to assess the differences between the single conditions (anatomical and experimental phases).
2) In my opinion in the Discussion section, it should be stressed the impact of the findings in several application fields. In fact, such an approach could be used in sports science, but also for rehabilitative purposes, and clinical settings in general to analyze several physiological signals.
Round 2
Reviewer 2 Report
Comments and Suggestions for Authors
I want to thank the Aurhors for addressing my concerns. In my opinion, the paper is improved and suitable for publication in the present form.
Author Response
We thank the reviewers for their efforts in reviewing our manuscript.